# Comparison of the Patency and Regenerative Potential of Biodegradable Vascular Prostheses of Different Polymer Compositions in an Ovine Model

**DOI:** 10.3390/ijms24108540

**Published:** 2023-05-10

**Authors:** Larisa V. Antonova, Viktoriia V. Sevostianova, Vladimir N. Silnikov, Evgeniya O. Krivkina, Elena A. Velikanova, Andrey V. Mironov, Amin R. Shabaev, Evgenia A. Senokosova, Mariam Yu. Khanova, Tatiana V. Glushkova, Tatiana N. Akentieva, Anna V. Sinitskaya, Victoria E. Markova, Daria K. Shishkova, Arseniy A. Lobov, Egor A. Repkin, Alexander D. Stepanov, Anton G. Kutikhin, Leonid S. Barbarash

**Affiliations:** 1Department of Experimental Medicine, Research Institute for Complex Issues of Cardiovascular Diseases, 6 Sosnovy Boulevard, Kemerovo 650002, Russia; sevovv@kemcardio.ru (V.V.S.); kriveo@kemcardio.ru (E.O.K.); veliea@kemcardio.ru (E.A.V.); miroav@kemcardio.ru (A.V.M.); shabar@kemcardio.ru (A.R.S.); senoea@kemcardio.ru (E.A.S.); hanomu@kemcardio.ru (M.Y.K.); glushtv@kemcardio.ru (T.V.G.); akentn@kemcardio.ru (T.N.A.); cepoav@kemcardio.ru (A.V.S.); markve@kemcardio.ru (V.E.M.); shidk@kemcardio.ru (D.K.S.); kytiag@kemcardio.ru (A.G.K.); reception@kemcardio.ru (L.S.B.); 2Laboratory of Organic Synthesis, Institute of Chemical Biology and Fundamental Medicine of the Siberian Branch of the Russian Academy of Sciences, Novosibirsk 630090, Russia; silnik@niboch.nsc.ru; 3Department of Regenerative Biomedicine, Research Institute of Cytology, 4 Tikhoretskiy Prospekt, St. Petersburg 194064, Russia; lobov@incras.ru; 4Centre for Molecular and Cell Technologies, St. Petersburg State University, Universitetskaya Embankment, 7/9, St. Petersburg 199034, Russia; st049553@student.spbu.ru; 5Institute of Medicine, Kemerovo State University, 6 Krasnaya Street, Kemerovo 650000, Russia; sasste334@gmail.com

**Keywords:** tissue-engineered vascular graft, biodegradable polymer, ovine model, carotid artery, antithrombotic coating, antibacterial coating, primary patency

## Abstract

The lack of suitable autologous grafts and the impossibility of using synthetic prostheses for small artery reconstruction make it necessary to develop alternative efficient vascular grafts. In this study, we fabricated an electrospun biodegradable poly(ε-caprolactone) (PCL) prosthesis and poly(3-hydroxybutyrate-co-3-hydroxyvalerate)/poly(ε-caprolactone) (PHBV/PCL) prosthesis loaded with iloprost (a prostacyclin analog) as an antithrombotic drug and cationic amphiphile with antibacterial activity. The prostheses were characterized in terms of their drug release, mechanical properties, and hemocompatibility. We then compared the long-term patency and remodeling features of PCL and PHBV/PCL prostheses in a sheep carotid artery interposition model. The research findings verified that the drug coating of both types of prostheses improved their hemocompatibility and tensile strength. The 6-month primary patency of the PCL/Ilo/A prostheses was 50%, while all PHBV/PCL/Ilo/A implants were occluded at the same time point. The PCL/Ilo/A prostheses were completely endothelialized, in contrast to the PHBV/PCL/Ilo/A conduits, which had no endothelial cells on the inner layer. The polymeric material of both prostheses degraded and was replaced with neotissue containing smooth-muscle cells; macrophages; proteins of the extracellular matrix such as type I, III, and IV collagens; and vasa vasorum. Thus, the biodegradable PCL/Ilo/A prostheses demonstrate better regenerative potential than PHBV/PCL-based implants and are more suitable for clinical use.

## 1. Introduction

Vascular tissue engineering is a field that can offer a method to develop new and efficient vascular grafts [1]. One of the main approaches of vascular tissue engineering is the creation of a highly patent vascular prosthesis that ensures the formation of tissue identical to the native vasculature. This prosthesis could be based on a tubular scaffold made from biocompatible biodegradable natural or synthetic polymers. High porosity is an important feature of the vascular prosthesis, since it provides an environment to promote cell migration from the blood stream and surrounding tissues into its wall, and supports the proliferation and differentiation of vascular cells. However, a prosthesis with this structure is susceptible to thrombosis and microbial contamination. Therefore, to prevent thrombus formation and infection after implantation, the surface of the tissue-engineered highly porous vascular prosthesis could be modified with antithrombotic and antibacterial drugs.

Nowadays, bacterial infections are widely recognized as some of the main global health issues. Microbial adhesion and colonization on the surfaces lead to the formation of biofilms with high resistance to host defense mechanisms and conventional antibiotic therapy [2,3]. Bacteria capable of forming biofilms and associated with implanted medical devices are common infection agents [4,5]. However, the conventional prophylactic administration of systemic antibiotics or even combinations thereof is not effective against biofilm-associated infections [6]. It has to do with the inability to determine a sufficient concentration of antibiotics at the site of biofilm formation [7]. Bacterial cells in biofilms are surrounded by a self-produced matrix that is resistant to the penetration of the antibiotic. In addition, biofilms contain large proportions of metabolic inactive persister cells, while antibiotics mostly target key bacterial transcriptional, translational, or metabolic pathways [8,9]. Therefore, bacterial biofilms are more resistant to antibiotics compared to free-form bacteria. Moreover, antibiotics are generally designed against free-form bacteria. Thus, the established minimum inhibitory concentrations for antibiotics and recommended doses may be insufficient to prevent the growth of bacterial biofilms [10].

The prevention of vascular prosthetic infection could be achieved via the introduction of antimicrobial agents into the structure of the prosthesis. This approach ensures the local delivery of the drug, which inhibits bacterial growth prior to biofilm formation [11,12,13,14,15]. Cationic surfactants (CSs) are promising antibacterial compounds for the treatment of antibiotic resistance bacterial infections. CSs are amphiphilic molecules composed of one or more hydrophilic positively charged groups and lipophilic moieties able to disrupt the transmembrane potential of the bacteria, causing leakage of the cytoplasmic contents and cell death [12]. Most of these compounds are highly active against both Gram-positive and Gram-negative bacteria, including antibiotic-resistant strains with selectivity toward bacterial cells over mammalian cells. CSs are able to significantly reduce the likelihood of developing bacterial resistance by affecting multiple targets in bacterial cells. Both their high stability and low cost of synthesis of CSs make them the most promising low molecular weight compounds for graft surface modification.

The other important requirement for tissue-engineered vascular prostheses is hemocompatibility. We have previously found that thrombotic occlusion of vascular prostheses implanted in the carotid artery of sheep may occur within the first hours or days after implantation, while the formation of new vascular tissue requires several months [16,17]. Thrombotic occlusion blocks the blood flow and affects the remodeling of the prosthesis wall, hindering its regeneration in situ. Highly thromboresistant material is particularly crucial for small-diameter vascular prostheses (less than 3 mm in diameter) due to the low blood flow in small blood vessels, which contributes to a high risk of thrombosis [18,19]. However, to date, there are no commercially available prostheses less than 3 mm in diameter, and the small-diameter vascular prostheses currently under development are still at the preclinical stage [20,21,22].

Antithrombotic coatings can be formed on the vascular prosthesis’ surface via the immobilization of drugs that inhibit platelet adhesion, activation, and aggregation, such as prostacyclin and its analogues. Prostacyclin analogues not only have antithrombotic properties but also exhibit an anti-inflammatory effect. Hence, modification with prostacyclin analogues can improve the thromboresistance of the prosthesis and stimulate vascular tissue regeneration by attracting the monocyte–macrophage lineage cells and reducing leukocyte adhesion and migration.

In a previous study, we modified the surface of a tissue-engineered vascular prosthesis made of poly-ε-caprolactone (PCL) with iloprost and a cationic amphiphile. The drug-loaded PCL prostheses exhibited 50% patency after long-term implantation in the ovine carotid artery, with polymer degradation synchronized with vascular tissue regeneration [23]. PCL is widely used for fabricating tissue-engineered vascular prostheses due to its high strength and very long biodegradation period. The main disadvantage of this polymer is its relatively hydrophobic character. Although a hydrophobic surface does not always lead to the desired cellular response, it can negatively affect the remodeling of the prosthesis after implantation [24].

We hypothesized that combining PCL with the highly biocompatible polymer poly(3-hydroxybutyrate-co-3-hydroxyvalerate) (PHBV) could improve the vascular prosthesis’ performance in vivo. PHBV is a biodegradable and biocompatible polyester that promotes cellular adhesion and proliferation. Electrospun vascular prostheses based on PHBV/PCL compositions have demonstrated excellent cytocompatibility [25].

In this study, we compared the mechanical properties, hemocompatibility, long-term patency, and tissue regeneration features of drug-loaded prostheses fabricated with PCL and PHBV/PCL in a sheep interposition carotid artery model.

## 2. Results

### 2.1. PVP Coating Provides Drug Attachment on the Prosthesis Surface

Mass spectrometry was performed to confirm the presence of iloprost and a cationic amphiphile on the surfaces of the PHBV/PCL/PVP/Ilo/A prostheses. An analysis of PBS samples after the incubation of the prostheses showed the lack of a peak at m/z 871.42256 for the cationic amphiphile C_41_H_84_Br_3_N_4_+ (Figure 1).

However, we observed peaks corresponding to phosphate salts of this ion, which were formed as a result of ion exchange with PBS. The spectrum also contained a peak at m/z 359,2000, which corresponded to the anion of iloprost (Appendix A).

The efficacy of the drug modification of the PCL/PVP/Ilo/A prostheses, as well as the kinetics of the drug release, were demonstrated in our previous study [23].

### 2.2. Drug Loading Does Not Diminish Mechanical Properties of Prostheses

The evaluation of the long-term patency of the developed vascular prostheses was carried out in a sheep model. Therefore, the study of the mechanical properties of grafts was conducted using the carotid artery of a sheep as a control.

First, we assessed the impact of PVP grafting on the mechanical properties of prostheses. Obtained via ionizing radiation, the PVP coating of the PCL prosthesis reduced its tensile strength by 1.6 times and increased the Young’s modulus by 1.4 times (*p* < 0.05, Table 1). However, the PVP modification of the PHBV/PCL prosthesis did not affect the strength but led to an increase in elastic modulus of 1.9 times (*p* < 0.05).

The results of the mechanical testing of the PVP-coated prostheses after the attachment of iloprost and the cationic amphiphile showed no change in the tensile strength for either type of prostheses. Moreover, we observed a decrease in the stiffness of the PHBV/PCL/PVP/Ilo/A samples compared to PHBV/PCL/PVP.

Both drug-modified prostheses had higher strength and rigidity compared to the native sheep carotid arteries.

### 2.3. Drug Loading Improves Hemocompatibility of the Prostheses

The study of the platelet aggregation activity did not reveal a difference between the unmodified PHBV/PCL and PCL prostheses. However, for the polymer prostheses, the values of this parameter were higher than for the platelet-rich plasma (PRP). PVP grafting on the inner surface of the prostheses contributed to a slight increase in platelet aggregation (Table 2). However, the subsequent complexation between the iloprost and cationic amphiphil with PVP resulted in a decrease in platelet aggregation on the surfaces of both types of prostheses by 6–7 times compared with that of intact PRP (*p* < 0.05, Table 2).

The hemolysis caused by contact of the blood with the surface of the prosthesis did not exceed the permissible 2% for all samples (Table 2). Moreover, there were no significant differences between the groups in terms of the degree of hemolysis.

The results of the platelet adhesion study showed that the number of platelets adhered on the surface of the PCL prosthesis was 3 times higher than in PHBV/PCL (Table 3).

Type II and type III platelets were predominantly observed on the surfaces of the unmodified samples. However, the PCL prosthesis had a 1.4-fold higher PDI than the PHBV/PCL (*p* < 0,05, Table 3). After the PVP modification of the surface, we found a 2-fold increase in the number of adherent platelets on the PHBV/PCL samples, as well as an increment of the proportion of type IV platelets on both types of prostheses (Table 3). Nevertheless, the PDI values did not change after PVP-coating the PCL and PHBV/PCL prostheses. Drug loading with the iloprost and cationic amphiphile led to a decrease numbers of adherent platelets on the PCL and PCL/PHBV samples by 1.3 and 1.5 times, respectively. Moreover, on the surface of the PCL/PHBV/PVP/Ilo/A prosthesis, the proportion of type III platelets was 2-fold lower than on the non-modified sample, and types I and II platelets prevailed over other types. In addition,, we observed an increase in the number of type III platelets on the PCL/PVP/Ilo/A. However, a significant reduction in PDI was noted for both types of drug-loaded prostheses (*p* < 0.05).

### 2.4. PCL/PVP/Ilo/A Prosthesis Demonstrates Better Long-Term Patency vs. PHBV/PCL/PVP/Ilo/A Prosthesis

PCL/PVP/Ilo/A and PHBV/PCL/PVP/Ilo/A biodegradable vascular prostheses were implanted into the carotid arteries of sheep for 6 months. Each animal received one graft. All animals survived until the end of the experiment. We examined the implanted vascular prostheses using Doppler ultrasound at days 1, 5, and 14 following implantation, and did not detect blood flow in the lumen of some of the prostheses (Appendix A). The primary patency rates of the PCL/PVP/Ilo/A and PHBV/PCL/PVP/Ilo/A prostheses were 83.3% and 67.7%, respectively. The Doppler ultrasound images of occluded prostheses 1 day after implantation corresponded to those obtained for the same prostheses at day 14 (Figure 2, Appendix A). The absence of blood flow in the same prostheses 1, 5, and 14 days after implantation indicated that these implants were thrombosed.

After 1 month of implantation, the primary patency rates were 83.3% and 67.7% in the PCL/PVP/Ilo/A group and the PHBV/PCL/PVP/Ilo/A group, respectively. By three months postimplantation, we observed a decrease in patency rates for both types of implants, with PCL/PVP/Ilo/A prostheses showing a patency rate of 50.0% and PHBV/PCL/PVP/Ilo/A showing a patency rate of 33.3%. At the end of the 6-month period, the patency of the PCL/PVP/Ilo/A prostheses remained at 50.0%, while all PHBV/PCL/PVP/Ilo/A implants had become thrombosed (Figure 2). 

### 2.5. Biodagradable Prostheses Support Arterial Regeneration In Situ

A macroscopic examination of the explanted PCL/PVP/Ilo/A prostheses revealed the integrity of the implant walls, the absence of aneurysms, and the growth of new tissue into the tubular scaffold (Figure 3).

Histological and immunofluorescent analyses of the explanted PCL/PVP/Ilo/A prostheses confirmed polymer resorption and revealed the formation of a three-layered vascular wall. The inner layer of the regenerated artery was neointima-lined with a continuous monolayer of mature endothelial cells that synthesized von Willebrand factor (Figure 4 and Figure 5A, Appendix A). The neointima consisted of smooth-muscle a-actin-stained cells, and types III and IV collagens localized predominantly in the newly formed basement membrane (Figure 4 and Figure 5A). The middle layer contained the remnants of the polymer scaffold, cells, collagen fibers, and numerous vasa vasorum (Figure 4 and Figure 5A). The outer layer resembled the adventitia of an artery and was composed of collagen fibers and a small number of cells.

Macroscopic evaluation of the explanted PHBV/PCL/PVP/Ilo/A prostheses also showed neotissue ingrowth, but an obturating thrombus was found in the lumen of all of these explants (Figure 5). Results of histological analysis and specific immunofluorescent staining demonstrated the absence of endothelial layer on the luminal surface. The polymer scaffold was replaced by cells and connective tissue (Figure 4 and Figure 5A). Smooth muscle cells and collagens type III and IV were observed in the middle part of the explant (Figure 5A). A layer similar to the tunica adventitia was formed outside the prosthesis. It consisted of connective tissue elements, fibroblast-like cells, macrophages, multinucleated giant cells and vasa vasorum. A small number of endothelial cells were found predominantly in the vasa vasorum (Figure 4 and Figure 5A). We did not observe any sings of acute inflammation or calcification in both types of explants.

The quantitative analysis showed no statistically significant differences in the fluorescence intensity of CD31 between PCL/PVP/Ilo/A (3.87 (2.46; 6.13) A.U.) and PHBV/PCL/PVP/Ilo/A (6.25 (3.57; 8.13) A.U.) (Figure 5B). The large amount of CD31 protein in the PHBV/PCL/PVP/Ilo/A samples was observed due to the increased number of blood vessels in the wall of the thrombosed prostheses. Additionally, the intensity levels of both CD31 and vWF in the wall of the native carotid artery were higher than for PCL/PVP/Ilo/A (*p* = 0.0161 and *p* = 0.0091, respectively) (Figure 5B).

The α-SMA staining of the vascular prostheses was determined in all layers (neointima, middle layer, and surrounding tissues) of the remodeled implants (Figure 5B). We did not observe any differences in the numbers of α-SMA-positive cells located in the middle layer of regenerated or native vessels among both groups of prostheses and the native coronary artery. However, a higher number of α-SMA-positive cells was observed in the neointima and middle layer of the PCL/PVP/Ilo/A compared with the surrounding tissue. At the same time, α-SMA expression in the surrounding tissues was found mainly in blood vessels (Figure 5B).

The quantitative analysis further indicated significantly higher amounts of collagen III in the walls of both PCL/PVP/Ilo/A and the carotid artery than for the PHBV/PCL/PVP/Ilo/A (*p* = 0.0377 and *p* = 0.0428, respectively) (Figure 5B). However, the native carotid artery displayed more collagen III in the adventitia compared with both polymeric prostheses. The anti-collagen IV staining revealed a higher amount of this protein in the carotid artery than in the prostheses. Meanwhile, we found no difference in the fluorescence intensity of collagen I between the prosthetic implants and native artery (Figure 5B).

The gene expression profiles in homogenates of the remodeled PHBV/PCL/PVP/Ilo/A prostheses were characterized by higher levels of mRNA-encoding anti-inflammatory cytokines (IL10, TGFB), as well as a molecule that promotes revascularization (CXCR4), compared with the contralateral native carotid artery of sheep (Figure 6). At the same time, the genetic landscape of PCL/PVP/Ilo/A vascular prostheses differed from that of the native carotid artery based on an increased level of mRNA of proteins involved in the inflammatory process (IL1B, CXCL8), endothelial reprogramming (NR2F2), and endothelial–mesenchymal transition (SNAI2) (Figure 6).

Next, we quantified the markers of platelets (CD41) and immune cells (CD68 and CD14) in polymeric prostheses and contralateral carotid arteries. All of these proteins were significantly over-represented in the prostheses, confirming their monocyte- and macrophage-mediated remodeling (Figure 7A–C). The PCL/PVP/Ilo/A and PHBV/PCL/PVP/Ilo/A implants did not show any significant differences in CD41, CD68, or CD14 amounts (Figure 7D–F). The proteomic profiling (i.e., ultra-high performance liquid chromatography–tandem mass spectrometry analysis) identified 182 proteins expressed in at least 4/6 prosthesis samples in at least one group (PCL/PVP/Ilo/A or PHBV/PCL/PVP/Ilo/A). The bioinformatics analysis found 17 proteins overexpressed in PCL/PVP/Ilo/A as compared with PHBV/PCL/PVP/Ilo/A prostheses, and 11 of these were involved in the actin rearrangements, suggesting the enhanced formation of a vascular smooth muscle layer in PCL/PVP/Ilo/A conduits (Table 4). Among the most notable were alpha smooth muscle actin (ACTA), smooth muscle myosin light chain kinase (MYLK), gamma-enteric smooth muscle actin (ACTH), calponin-1 (CNN1), transgelin/SM22α (TAGL), and caldesmon (CALD1).

## 3. Discussion

Biodegradable vascular prostheses are promising alternatives to the currently used small-diameter prostheses. Implanted porous prostheses made of biodegradable polymers can be populated by cells migrating from the bloodstream and surrounding tissues, replacing the polymer material and forming a new blood vessel [26]. However, the use of the materials with high porosity for vascular prosthesis fabrication could cause thrombosis and bacterial contamination. This must be taken into account in the development of implantable medical devices for cardiovascular surgery. Surface modification with antithrombotic agents is a common method used to prevent the thrombosis of implanted vascular prostheses in the early postoperative period [27]. Additionally, introducing antibacterial agents to the biodegradable implant can reduce the risk of bacterial contamination during the entire period of the existence of the prosthesis [28].

Previously, we developed PCL and PHBV/PCL scaffolds modified with iloprost and a cationic amphiphile to improve their hemocompatibility and antibacterial properties, respectively. The drugs were attached to the scaffold’s surface using a PVP hydrogel. The biocompatibility assessment of these materials showed that the PVP coating improved the adhesion and viability of the endothelial cells by 50% [25]. The proportion of proliferating cells on the scaffolds coated with drugs reached 100%, which was significantly higher than in the samples without the drugs.

Moreover, we confirmed the high bacteriostatic activity of the cationic amphiphile attached to the surfaces of the scaffolds against the following strains of microorganisms: Klebsiella pneumoniae spp. ozaenae, Escherichia coli, Staphylococcus aureus, Proteus mirabillis, and Pseudomonas aeruginosa [29]. These microorganisms are the most common causes of infectious complications in cardiac surgery. Electrospun scaffolds modified with cationic amphiphile exhibited a local inhibitory effect on the bacteria. The difference in the polymer composition of the scaffolds with the cationic amphiphile did not affect their bacteriostatic properties. Moreover, we found that scaffolds loaded with the cationic amphiphile at a concentration of 0.25 mg/mL exhibited integrated antibacterial activity and hemocompatibility in vitro [29]. As a result, they were selected for preclinical testing in an animal model.

The hydrophobicity of PCL limits its use for vascular prostheses due to its insufficient hemocompatibility, while PHBV, although being more biocompatible, has poor mechanical strength [30,31,32]. To address this, we combined PCL and PHBV to compensate for the limitations of each polymer. In the present study, we evaluated the properties of PCL and PHBV/PCL biodegradable prostheses loaded with iloprost and the cationic amphiphile.

The drug modification significantly improved the hemocompatibility and mechanical strength of both types of prostheses. However, the outcomes of the implantation of PCL/PVP/Ilo/A and PHBV/PCL/PVP/Ilo/A vascular prostheses in the carotid arteries of sheep differed from each other. Due to thrombosis, the PHBV/PCL/PVP/Ilo/A graft had worse primary patency compared with PCL/PVP/Ilo/A samples at different time points after implantation. This was likely due to the greater stiffness of the PHBV/PCL/PVP/Ilo/A prosthesis, since a large stiffness mismatch between the native artery and vascular prosthesis may affect the hemodynamics in the anastomosis region and cause thrombus formation [33]. In turn, the PCL prostheses did not undergo extensive thrombosis. The six-month patency rate of these implants was 50%. An earlier study showed a 78% (7/9) primary patency rate for PCL grafts implanted into the carotid arteries of pigs for 4 weeks [34]. In another work, 100% (3/3) of PCL vascular scaffolds failed at 10 days after ovine carotid artery replacement due to blood clots [35]. However, the long-term patency of these prostheses was achieved by seeding endothelial cells and smooth muscle cells onto the PCL vascular scaffolds using a bioreactor [35,36]. Additionally, Ye et al. showed that PCL grafts with conjugated heparin were 100% patent after implantation into a dog’s femoral artery for 4 weeks [37]. 

Vascular prostheses containing PHBV have not been as extensively studied as PCL prostheses. Some in vitro studies have demonstrated the good hemocompatibility of the porous tubular scaffolds based on PHBV, and excellent cell attachment and growth on their surfaces [38,39]. In a small animal model, PHBV/PCL prostheses exhibited high patency and efficient remodeling [40]. However, in our in vivo study in the sheep model, we found a dramatic decrease in the thromboresistance of vascular prostheses as a result of adding PHBV to PCL. The PHBV/PCL/PVP/Ilo/A prostheses showed poor patency despite their antithrombotic surface modification, likely due to their high rigidity. On the other hand, the coating of a polyvinylpyrrolidone (PVP) hydrogel containing iloprost on the surfaces of the PCL prostheses resulted in improved patency. The PVP probably reduced the PCL’s hydrophobicity, and the lower rigidity of these prostheses provided a satisfactory compliance mismatch between the host artery and vascular substitutes.

In our previous study, we evaluated the degradation rate of PCL and PHBV/PCL electrospun samples in vivo with subcutaneous implantation in rats [41]. The results showed a slow degradation rate for both polymer compositions, with incomplete degradation observed 12 months after subcutaneous implantation. However, prostheses implanted in the carotid arteries of sheep undergo much faster resorption. Within 3.5 months, we observed the formation of aneurysms in the walls of the prostheses due to the rapid bioresorption of the polymer material [42]. Other scientific groups announced the accelerated resorption of polymer products implanted in sheep in parallel with our findings in 2020 [26,43]. The authors attributed this to the unique characteristics of the sheep metabolism. The rate of biodegradation of a polymeric implant depends not only on the properties of the polymer, but also on the shape and morphology of the implant being tested, as well as the animal model used for testing. 

The histology and immunofluorescent analysis showed in situ endothelialization on the inner surfaces of PCL/PVP/Ilo/A prostheses. High expression levels of transcripts related to endothelial differentiation, proteins of endothelial reprogramming (NR2F2), and endothelial–mesenchymal transition (SNAI2) were observed in the remodeled prostheses, which also confirmed the endothelium formation. In contrast, we did not find any signs of endothelialization on the inner surfaces of PHBV/PCL/PVP/Ilo/A implants, but we noticed a large number of neoformed blood vessels in the middle layer of these prostheses and high gene expression levels of receptor CXCR4 involved in the revascularization process. These findings indicate enhanced angiogenesis in the remodeled walls of PHBV/PCL/PVP/Ilo/A implants. Interestingly, in PCL/PVP/Ilo/A prostheses, the CXCR4 gene expression was 2.13 times higher than in the PHBV/PCL/PVP/Ilo/A prostheses. CXCR4 is the chemokine receptor for CXCL12 (also known as SDF-1), which is expressed by various cell types, including macrophages and T cells. This receptor plays an important role in the homing of progenitor and immune cells [44]. We believe that the increase in the expression of genes associated with angiogenesis was not dependent on the type of polymer or composition but on the intensity of migration of cells expressing these genes, specifically macrophages involved in polymer bioresorption, into the prosthesis wall. It is likely that in patent PCL/PVP/Ilo/A prostheses, the infiltration of various cell types into the wall occurs from both the surrounding tissues and the bloodstream, while in thrombosed PHBV/PCL/PVP/Ilo/A prostheses, cell migration into the wall is only possible from the surrounding tissues. The degree of vascularization of the prosthesis wall could depend on the number of migrated cells.

The upregulation of cytokines IL10 and TGFB in the remodeled PHBV/PCL/PVP/Ilo/A prostheses can also be associated with enhanced new tissue ingrowth due to the migration of cells into the prosthesis wall from the surrounding tissues [45,46].

The remodeling of both types of protheses was confirmed by the presence of CD41, CD68, and CD14 cells. Neotissue had formed and replaced polymeric material within the walls of both prostheses. The PCL/PVP/Ilo/A implants were characterized by the upregulation of IL1B and CXCL8, which are involved in the inflammatory process. The increased expression of inflammatory transcripts may be associated with the colonization of prostheses by cells of the monocyte–macrophage linage, which play a key role in the biodegradation of polymeric material and implant remodeling [47]. Additionally, only PCL/PVP/Ilo/A prostheses showed enhanced formation of a smooth muscle cell layer, as demonstrated by the immunofluorescent staining and proteomic analysis.

Thus, despite the greater hydrophobicity of the PCL material compared to the composite based on a mixture of PCL and PHBV, the prosthesis made of PCL alone was less prone to thrombosis, and it underwent endothelialization and smooth muscle layer formation and maintained its long-term patency. Regarding the extensive thrombosis of PHBV/PCL/PVP/Ilo/A prostheses, the obtained data do not allow an unambiguous determination of whether early thrombosis inhibited the endothelialization of these conduits or if the lack of endothelialization caused thrombosis.

## 4. Materials and Methods

### 4.1. Preparation of Biodegradable Vascular Prostheses

Two different types of vascular prostheses with an inner diameter of 4 mm were fabricated via an electrospinning method using a NANON-01A electrospinning setup (MECC Co., Ltd., Fukuoka, Japan). The first type of prosthesis was made from a 12% poly(ε-caprolactone) (PCL) solution in 1,1,1,3,3,3-hexafluoro-2-propanol (HFP). The second type was produced by mixing 12% PCL and 2% poly(3-hydroxybutyrate-co-3-hydroxyvalerate) solutions in HFP in a 1:2 *v*/*v* ratio. The electrospinning was carried out using a cylindrical stainless steel rotating collector (ⵁ = 4 mm), with the process parameters specified in Table 5.

### 4.2. Drug Surface Modification

The inner surfaces of the prostheses were coated with a polyvinylpyrrolidone (PVP) hydrogel via radiation-induced grafting polymerization [48].

The electrospun PCL and PHBV/PCL prostheses were first immersed in a 5% ethanol solution of PVP (K 90; PanReac AppliChem, Darmstadt, Germany) for 30 min and then removed from the solution and dried at 23 °C for 24 h. The PVP grafting onto the surface was carried out using ionizing radiation under an argon atmosphere with an absorbed dose of 15 kGy, using a 50 kW ILU-10 accelerator (5 MeV, G.I. Budker INP SB RAS, Novosibirsk, Russia). The samples were washed with sterile distilled water for 60 min. The PVP coating on the inner surfaces of the prostheses was evaluated via Fourier transform infrared (FT-IR) spectroscopy on a Bruker Vertex 80 V FT-IR spectrometer (Bruker Optik GmbH, Ettlingen, Germany) equipped with an ATR accessory (Bruker Optik GmbH, Ettlingen, Germany) in the spectral range of 4000–500 cm^−1^.

The surfaces of the prostheses were further modified by attaching the antithrombotic drug iloprost and the antibacterial agent 1,5-bis-(4-tetradecyl-1,4-diazoniabicyclo [2.2.2]octan-1-yl)pentane tetrabromide. Iloprost is a synthetic analog of prostacyclin, which inhibits platelet release reactions, adhesion, and aggregation. It also restores disturbed microcirculation by inducing vasodilation, inhibiting platelet activation, and activating fibrinolysis, and inhibits the adhesion and migration of leukocytes after endothelial injury. The cationic amphiphile 1,5-bis-(4-tetradecyl-1,4-diazoniabicyclo [2.2.2]octan-1-yl)pentane tetrabromide has ribonuclease activity that defines its high antibacterial and antiviral activity [49,50,51].

In this study, 1,5-bis-(4-tetradecyl-1,4-diazoniabicyclo [2.2.2]octan-1-yl)pentane tetrabromide was synthetized as previously described (Appendix A) [52]. The data on the spectra of the obtained compound properly coincided with the literature data [52].

Iloprost and 1,5-bis-(4-tetradecyl-1,4-diazoniabicyclo [2.2.2]octan-1-yl)pentane tetrabromide were attached to the protheses’ surfaces through complexation with the PVP coating layer. To achieve this, the protheses were introduced to a sterile aqueous solution containing 0.25 mg/mL of 1,5-bis-(4-tetradecyl-1,4-diazoniabicyclo [2.2.2]octan-1-yl)pentane tetrabromide (A) and 0.2 mg/mL of iloprost (Ilo) for 30 min. Next, the PCL/PVP/Ilo/A and PHBV/PCL/PVP/Ilo/A samples were air-dried under sterile conditions and placed in sterile containers.

Unmodified PCL and PHBV/PCL samples, as well as protheses coated with PVP without incorporated drugs (PCL/PVP and PHBV/PCL/PVP), were used as controls. All samples are presented in Table 6.

### 4.3. Determination of Drug Loading

To verify the drug surface modification of PHBV/PCL/PVP/Ilo/A protheses, mass spectrometry was employed. Each sample of 1 mm^2^ was incubated in 1 mL of sterile phosphate–saline buffer (PBS, 70011-036, Gibco life technologies, NY, USA) containing 0.05% Tween-20 (P1379, Sigma-Aldrich, Saint Louis, MO, USA) at 37 °C for 24 h. The spectra of the PBS solution were determined by an Agilent 1100 Series Liquid Chromatograph–Mass Selective Detector (Agilent Technologies, Santa Clara, CA, USA) equipped with both electrospray ionization and atmospheric pressure chemical ionization modes.

### 4.4. Mechanical Testing

Samples were cut from the vascular prostheses using a ZCP 020 manual cutting press (Zwick/Roell, Ulm, Germany). The mechanical properties of the samples were evaluated through uniaxial tensile tests using a Z-series universal testing machine (Zwick/Roell, Ulm, Germany) with a 50 N load cell at a strain rate of 50 mm/min. The stress–strain curves were used to obtain the ultimate tensile strength, elongation at break, and Young´s modulus. Intact ovine carotid arteries were used as control samples.

### 4.5. Hemocompatibility Studies

The hemocompatibility studies of the drug-loaded and non-drug-loaded prostheses were carried out according to ISO10993.4. Blood samples from healthy human volunteers were collected in tubes containing 3.8% sodium citrate at a blood/sodium citrate ratio of 9:1.

#### 4.5.1. Hemolysis Test

Prosthetic samples of 25 mm^2^ (n = 5 per group) were placed in vails containing 10 mL of 0.9% saline solution (P010p, Paneco, Moscow, Russia) and incubated at 37 °C for 120 min. Here, 0.9% saline solution and distilled water were used as the negative and positive controls, respectively. Next, 200 µL of citrated blood was added to each vial, and the samples were incubated at 37 °C for 60 min. Then, the samples were transferred from the vials to tubes and centrifuged at 789 g for 10 min. The absorbance intensity of the supernatant was measured at 540 nm using a GENESYS 6 spectrophotometer (Thermo Fisher Scientific, Madison, WI, USA).

The percentage of hemolysis was calculated using Equation (1) [52,53]:(1)$$H(\%)=\frac{D_{t}-D_{\mathrm{ne}}}{D_{\mathrm{pe}}-D_{\mathrm{ne}}}\times 100$$
where D_t_ is the absorbance of the sample; D_ne_ is the absorbance of the negative control (citrate blood sample with 0.9% saline solution); D_pe_ is the absorbance of the positive control (citrate blood sample with distilled water).

#### 4.5.2. Platelet Aggregation Test

To obtain PRP, the citrate blood was centrifuged at 101 g for 10 min. Platelet-poor plasma (PPP) was obtained via the re-centrifugation of PRP at 101 g for 10 min. Each prosthetic sample was then incubated with 250 µL of PRP for 3 min, followed by the addition of 25 µL of 0.025 M calcium chloride (HB-2201-FG, Hart biologicals, Hartlepool, UK). Spontaneous platelet aggregation was measured using an APACT 4004 semi-automated platelet aggregometer (LABiTec, Ahrensburg, Germany). PRP and PPP were used as positive and negative controls, respectively.

#### 4.5.3. Platelet Adhesion Test

Prosthetic samples of 5 mm^2^ (n = 2 per group) were incubated with 300 µL of PRP obtained as described above at 37 °C for 2 h and then washed with PBS to remove non-adhered components of RPR. The samples were fixed in 2% glutaraldehyde for 24 h; washed with PBS; dehydrated in consecutive 30%, 50%, 70%, 90%, and 100% ethanol solutions for 15 min each; and air-dried. Next, all samples were mounted onto stubs with double-sided carbon adhesive tape and sputter-coated (10 nm thick) with Au/Pd using a Leica EM ACE200 Sputtering System (Leica Mikrosysteme GmbH, Wien, Austria). The surfaces of materials were visualized with an S-3400N scanning electron microscope (Hitachi, Tokyo, Japan). The numbers of adherent platelets were counted in 8 randomly selected fields for each sample and presented as numbers of platelets per 1 mm^2^. 

The observed platelets were categorized according to their morphology: type I—discoid shape, no deformation; type II—pseudopodia sticking out (early pseudopodial); type III—irregular shape with pronounced pseudopodia, aggregation of platelets (intermediate pseudopodial); type IV—flat platelets with cytoplasm spreading among pseudopodia (late pseudopodial); type V—cytoplasm fully spread, the shape of pseudopodia cannot be seen clearly.

The platelet deformation index was calculated as follows (Equation (2)) [52,53,54]:(2)$$\mathrm{PDI}=\frac{\mathrm{PI}+\mathrm{PII}\times 2+\mathrm{PIII}\times 3+\mathrm{PIV}\times 4+\mathrm{PV}\times 5}{\mathrm{Ptotal}}$$
where PI is the number of type I platelets; PII is the number of type II platelets; PIII is the number of type III platelets; PVI is the number of type IV platelets; PV is the number of type V platelets; Ptotal is the total number of platelets.

### 4.6. Implantation of the Prosthesis

In this study, 12 female Edilbay sheep weighing 42–45 kg were utilized. The study protocol was approved by the Local Ethical Committee of the Research Institute for Complex Issues of Cardiovascular Diseases (protocol number 15, 11 September 2018, Kemerovo, Russia). All animal experiments were conducted following the European Convention for the Protection of Vertebrate Animals (Strasbourg, 1986).

Vascular prostheses of PCL/PVP/Ilo/A and PHBV/PCL/PVP/Ilo/A with an inner diameter of 4 mm and length of 40 mm were implanted unilaterally in the sheep carotid artery (n = 6 per group).

All sheep were premedicated with xylazine in the dose range of 0.05–0.25 mL per 10 kg and with 1 mg of intramuscular atropine (Moscow Endocrine Plant, Moscow, Russia). Anesthesia was induced with 5–7 mg/kg of propofol, followed by an intravenous infusion of 0.5–0.6 mg/kg of atracurium besylate (Ridelat-C, Sotex, Moscow, Russia). The tracheal intubation was performed using a 9.0 endotracheal tube, and anesthesia was maintained with sevoflurane at 2–4 vol% and with a continuous infusion of atracurium besylate (0.3–0.6 mg/kg/h). Continuous monitoring of the arterial blood pressure, heart rate, and oxygen saturation was performed.

Following the intravenous administration of heparin (5000 IU, Moscow Endocrine Plant, Moscow, Russia), the carotid artery was clamped and a 4 cm segment was excised. Vascular prostheses were implanted with two end-to-end anastomoses using a 6-0 Prolene suture (Ethicon, Somerville, NJ, USA). Hemostasis was verified, and the wound was closed using a standard surgical technique using a 2-0 Vicril suture (Ethicon, Somerville, NJ, USA). All animals received a subcutaneous injection of enoxaparin sodium at 4000 anti-Xa IU/0.4 mL and were extubated.

The postoperative care included injections of 1.5 g of cefuroxime intramuscularly 3 times daily, 4000 anti-Xa IU/0.4 mL of enoxaparin sodium subcutaneously for 5 days, as well as 75 mg of clopidogrel (Krka-Rus, Istra, Russia) orally once a day and 5000 IU of heparin sodium subcutaneously twice daily for 30 days.

The patency of the prostheses was monitored using Doppler ultrasonography on days 1, 5, and 14, and at 1 month and every 3 months thereafter, up to 6 months. After 6 months postsurgery, the prostheses and segments of intact carotid artery were explanted and divided into 5 equal parts.

### 4.7. Histology and Immunofluorescence Examination

The explanted samples were fixed in 10% neutral phosphate-buffered formalin at 4 °C for 24 h. Each sample was then rinsed in running tap water, dehydrated in isopropyl alcohol IsoPrep (BioVitrum, St. Petersburg, Russia), impregnated three times in Gistomix paraffin (BioVitrum, St. Petersburg, Russia) at 56 °C for 60 min, and embedded in a paraffin block. Next, 8-μm-thick sections were cut using a HM-325 microtome (Thermo Scientific, Waltham, MA, USA) and dried overnight at 37 °C. After complete drying, the samples were deparaffinized three times in o-xylene for 2 min and dehydrated three times in 96% ethanol for 2 min. Next, the deparaffinized sections were stained with hematoxylin and eosin and Van Gieson or Alizarin Red S using conventional staining protocols. The stained samples were examined under an AXIO Imager A1 light microscope (Carl Zeiss, Goettingen, Germany).

For the immunofluorescence study, the explanted segments of prostheses and carotid arteries were frozen at −140 °C.

The 8-µm-thick sections were cut using CryoStarNX50 cryotome (Thermo Shandon Limited, Runcorn, UK). Each section was mounted onto a clean glass slide and fixed by immersion in 4% paraformaldehyde for 10 min. To stain the intracellular markers, the sections were permeabilized with PBS containing 0.1% Triton X100 (X100, Sigma-Aldrich, Burlington, MA, USA) for 15 min and blocked with 1% BSA for 1 h at room temperature. Next, the samples were incubated with primary antibodies at 4 °C overnight in the following combinations: (i) CD31 Rabbit Monoclonal Antibody (ab28364, Abcam, Cambridge, Great Britain)/α-smooth muscle actin (α-SMA) Mouse Monoclonal Antibody (ab7817, Abcam, Cambridge, Great Britain); (ii) von Willebrand Factor (vWF) Rabbit Monoclonal Antibody (ab6994, Abcam, Cambridge, Great Britain); (iii) Collagen IV Rabbit Monoclonal Antibody (ab6586, Abcam, Cambridge, Great Britain)/Collagen I Mouse Monoclonal Antibody (ab23446, Abcam, Cambridge, Great Britain); (iv) Collagen III Rabbit Monoclonal Antibody (NB600-594, Novus Biologicals, Centennial, CO, USA). The sections were then labeled with Donkey Anti-Rabbit IgG Secondary Antibody, Alexa Fluor 488 conjugate (A21206, Thermo Fisher, Waltham, MA, USA) or Donkey Anti-Mouse IgG Secondary Antibody, and Alexa Fluor 555-conjugate (A31570, Thermo Fisher, Waltham, MA, USA) for 1 h at room temperature. After each step of staining, the sections were washed with 0.1% Tween-20 (P9416, Sigma-Aldrich, Burlington, MA, USA) in PBS. To remove the autofluorescence, the samples were treated with an autofluorescence eliminator reagent (2160, Millipore, Burlington, MA, USA). The nuclei were counterstained with 10 μg/mL of DAPI (D9542, Sigma-Aldrich, Burlington, MA, USA) for 30 min. Finally, the stained sections were mounted under a coverslip using ProLong mounting medium (P36930, Thermo Fisher, Waltham, MA, USA). An ovine carotid artery was used as the control. The samples were examined with a scanning laser microscope confocal microscope LSM 700 (Carl Zeiss, Oberkochen, Germany). A semi-quantitative analysis of the images was performed in ImageJ software (National Institutes of Health, Bethesda, MD, USA).

### 4.8. Analysis of Gene Expression Profile

The samples of the harvested prostheses and contralateral arteries were homogenized in Trizol Reagent (15596018, Thermo Fisher Scientific, Waltham, MA, USA), and RNA extraction was performed following the manufacturer’s protocol (FastPrep-24 instrument and Lysing Matrix S, 116925050-CF, MP Bio-medicals, Irvine, CA, USA). The gene expression level was measured by quantitative reverse transcription polymerase reaction using a High-Capacity cDNA Reverse Transcription Kit (4368814, Thermo Fisher Scientific, Waltham, MA, USA). The primers were synthesized by Evrogen JSC (Moscow, Russia) on an ABI 3900 High-Throughput DNA Synthesizer (Thermo Fisher Scientific, Waltham, MA, USA). For each sample, a 20 μL reaction mixture was prepared with 5 µL of the PowerUpTM SYBR^®^ Green Master Mix (Applied Biosystems, Waltham, MA, USA), a mix of forward and reverse primers at a final concentration of 500 nM, and 10 ng of cDNA. Regarding the analyzed samples, five standards with two-fold dilution reaction mixture without cDNA as a negative control were placed in optical 96-well reaction plates (PCR-96-LP-FLT-C, Axygen, New York, NY, USA), and qPCR was performed on a CFX96 Touch instrument (BioRad, Hercules, CA, USA). All samples and negative controls were assayed in triplicate. Amplification was carried out under the following conditions: 50 °C for 2 min, 95 °C for 2 min, and 40 cycles of 95 °C for 15 s and 60 °C for 1 min. The qPCR results were normalized using three reference genes ACTB, GAPDH, and B2M (Table 7). The relative gene expression was calculated using the comparative (2^−ΔΔCt^) method, and expressed as a fold change relative to the control samples. Alternatively, the relative levels of gene expression in the lysate of endothelial cells of remodeled prostheses were expressed as a value relative to the housekeeping genes (ACTB, GAPDH, B2M). The adjusted *p*-values were presented in a heat map, where blue, gray, and red colors reflected fold changes ≤ 0.50, 0.51–1.99, and ≥2.00, respectively.

### 4.9. Western Blotting and Proteomic Analysis

For the protein assays, distal segments of PHBV/PCL/PVP/Ilo/A and PCL/PVP/Ilo/A prostheses (n = 6 per each group and n = 12 in total) and intact contralateral carotid arteries (n = 12) were flushed with physiological saline (Hematek, Tver, Russia) and homogenized (FastPrep-24 5G, MP Biomedicals, San Diego, CA, USA; Lysing Matrix S Tubes, 116925050-CF, MP Biomedicals, San Diego, CA, USA) in T-PER buffer (78510, Thermo Fisher Scientific, Waltham, MA, USA) supplied with the Halt protease and phosphatase inhibitor cocktail (78444, Thermo Fisher Scientific, Waltham, MA, USA) according to the manufacturer’s protocol. Upon the initial centrifugation at 14,000× *g* (Microfuge 20R, Beckman Coulter, Brea, CA, USA) for 10 min, the supernatant was additionally centrifuged at 200,000× *g* (Optima MAX-XP, Beckman Coulter, Brea, CA, USA) for 1 h to sediment insoluble ECM proteins. The quantification of the total protein was conducted using a BCA Protein Assay Kit (23227, Thermo Fisher Scientific, Waltham, MA, USA) and Multiskan Sky microplate spectrophotometer (Thermo Fisher Scientific, Waltham, MA, USA) in accordance with the manufacturer’s protocol. For the bioinformatics analysis of the proteomic profiling data, we used our previously published [23] dataset deposited to the ProteomeXchange Consortium via the PRIDE partner repository [55] with the dataset identifier PXD036520. A reproducible code for the data analysis is available from https://github.com/ArseniyLobov/Proteomic-profiling-of-grafts-implanted-into-the-ovine-artery (accessed on 3 October 2022). In the initial paper [23], we carried out a pairwise comparison of tissue-engineered vascular grafts and intact contralateral carotid arteries, whereas here we employed this dataset to compare PHBV/PCL/PVP/Ilo/A and PCL/PVP/Ilo/A prostheses.

Equal amounts of protein (25 μg per sample) were mixed with OrangeMark sample buffer (K-023, Molecular Wings, Kemerovo, Russia) in a 6:1 ratio, denatured at 99 °C for 5 min, and then loaded on a 15-well 1.5 mm NuPAGE 4–12% Bis-Tris protein gel plate (NP0336BOX, Thermo Fisher Scientific, Waltham, MA, USA). A Chameleon Duo Pre-Stained Protein Ladder (928–60000, LI-COR Biosciences, Lincoln, NE, USA) was loaded as a molecular weight marker. The proteins were separated using sodium dodecyl sulphate–polyacrylamide gel electrophoresis (SDS-PAGE) at 150 V for 2 h using G-RUN MES running buffer (K-021, Molecular Wings, Kemerovo, Russia), G-NOOOX antioxidant (K-027, Molecular Wings, Kemerovo, Russia), and an XCell SureLock Mini-Cell vertical mini-protein gel electrophoresis system (EI0001, Thermo Fisher Scientific, Waltham, MA, USA). Protein transfer was performed using nitrocellulose transfer stacks (IB23001, Thermo Fisher Scientific, Waltham, MA, USA) and an iBlot 2 Gel Transfer Device (Thermo Fisher Scientific, Waltham, MA, USA) according to the manufacturer’s protocols using a standard transfer mode for 30–150 kDa proteins (P0—20 V for 1 min, 23 V for 4 min, and 25 V for 2 min). Nitrocellulose membranes were then incubated in Block’n’Boost! solution (K-028, Molecular Wings, Kemerovo, Russia) for 1 h to prevent non-specific binding.

The blots were probed with mouse monoclonal antibodies to CD41/61 (1:500, MCA1095GA, Bio-Rad, Hercules, CA, USA), CD68 (1:500, M0718, Agilent Technologies, Santa Clara, CA, USA), and CD14 (1:500, MA5-28287, Thermo Fisher Scientific, Waltham, MA, USA). IRDye 680RD-conjugated goat anti-mouse IgG secondary antibody (926-68070, LI-COR Biosciences, Lincoln, NE, USA) was used at a 1:1000 dilution. Incubation with the antibodies was performed using Block’n’Boost! solution (K-028, Molecular Wings, Kemerovo, Russia), iBind Flex Cards (SLF2010, Thermo Fisher Scientific, Waltham, MA, USA), and a Bind Flex Western Device (Thermo Fisher Scientific, Waltham, MA, USA) according to the manufacturer’s protocols. Fluorescent detection was performed using an Odyssey XF imaging system (LI-COR Biosciences, Lincoln, NE, USA) at a 700 nm channel (685 nm excitation and 730 nm emission). The total protein normalization was conducted after the fluorescent detection using 0.1% Fast Green FCF (F8130, Solarbio Life Sciences, Beijing, China) dissolved in 30% methanol (8.05.00186, ChemExpress, Ufa, Russia) and 7% acetic acid (61-75, EKOS-1, Moscow, Russia). The staining of the nitrocellulose membrane with 0.1% Fast Green FCF (F8130, Solarbio Life Sciences, Beijing, China) for 10 min was followed by destaining in 30% methanol (8.05.00186, ChemExpress, Ufa, Russia) and 7% acetic acid (61-75, EKOS-1, Moscow, Russia) for another 10 min and in two changes of deionized water (2 min per each). The total protein visualization was performed using an Odyssey XF imaging system (LI-COR Biosciences, Lincoln, NE, USA) in a 600 nm channel (520 nm excitation and 600 nm emission). Densitometry was performed using ImageJ software (National Institutes of Health, Bethesda, MD, USA) using the standard algorithm (consecutive selection and plotting of the lanes with the measurement of the peak area) and with subsequent adjustment of the specific signal from the antibody to the level of the total protein transferred to a nitrocellulose membrane. Th statistical analysis was performed by employing the total protein-adjusted specific staining values.

### 4.10. Statistical Analysis

The data analyses were performed using GraphPad Prism 7.0 software. The normality of the data distribution was tested with the Kolmogorov–Smirnov test. The experimental data are presented as median and interquartile ranges (25–75%). Significant differences at the *p* < 0.05 level between more than two groups were tested using a Kruskal–Wallis H test with false discovery rate (FDR) correction for multiple comparisons or with a Dunn post-test method.

## 5. Conclusions

Based on the data obtained from our study, it can be inferred that the combination of PCL with PHBV does not improve the in vivo performance of small-diameter biodegradable prostheses based on PCL prostheses that were modified with antithrombotic and antibacterial coatings. The prostheses containing PHBV experienced extensive thrombosis, resulting in remodeling of the implant walls with the formation of structureless tissue. Conversely, patent prostheses made solely of PCL demonstrated satisfactory long-term patency and gradual degradation, eventually being replaced by newly formed three-layer tissue similar to vascular tissue. Our study has demonstrated the regenerative potential of biodegradable prostheses in a large animal model. However, the further improvement of the thromboresistance of the PCL prostheses is needed to enhance their in vivo performance. Additionally, further research is necessary to study the subpopulation of macrophages and their impact on the tissue-forming process to gain a better understanding of the vascular regeneration mechanisms. 

## Figures and Tables

**Figure 1 ijms-24-08540-f001:**
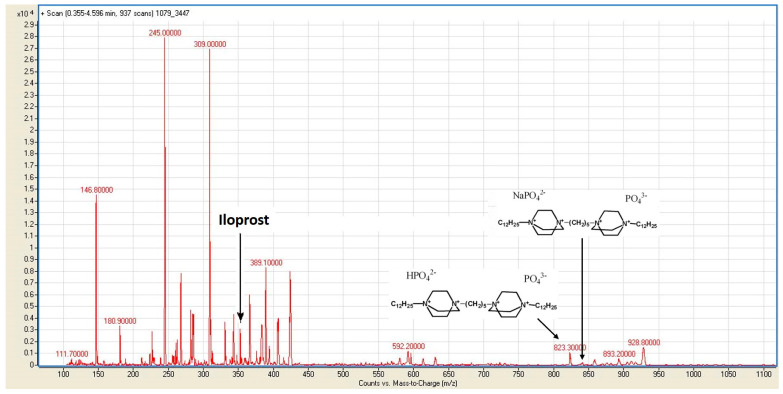
A mass-spectrometry analysis of PBS samples after incubation with PHBV/PCL/PVP/Ilo/A prostheses. The spectrum shows peaks corresponding to iloprost (m/z 359,2000) and phosphate salts of the cationic amphiphiles C_41_H_84_N_4_NaO_8_P_2_^−^ (m/z 845,3000) and C_41_H_85_N_4_O_8_P_2_^−^ (m/z 823,3000).

**Figure 2 ijms-24-08540-f002:**
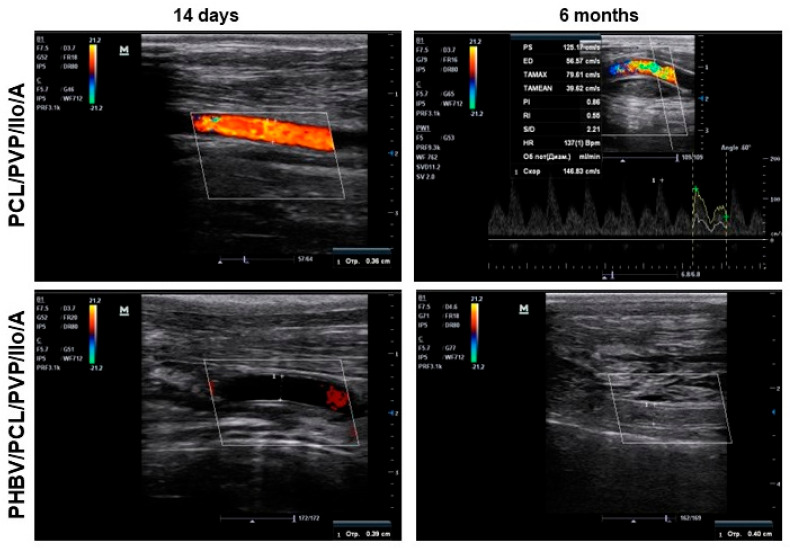
Representative ultrasound Doppler images of PCL/PVP/Ilo/A and PHBV/PCL/PVP/Ilo/A vascular prostheses in the ovine carotid artery at both 14 days and 6 months postsurgery.

**Figure 3 ijms-24-08540-f003:**
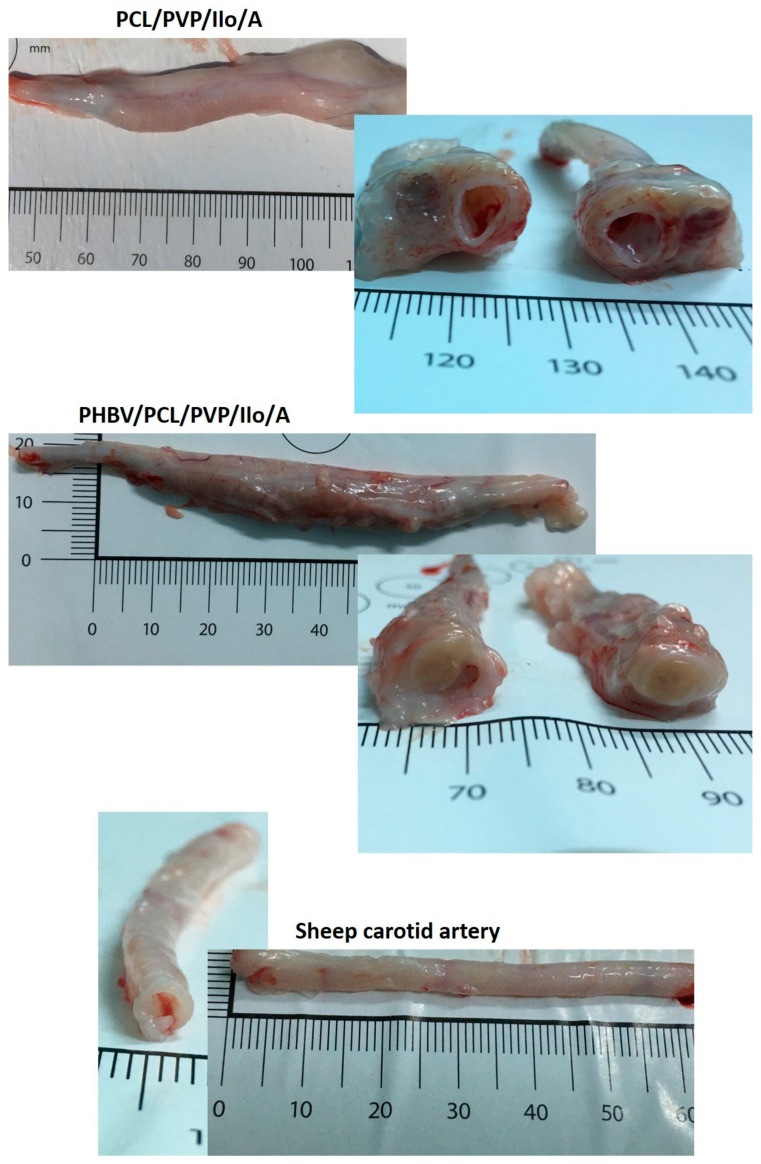
Macroscopic examination of explanted vascular PCL/PVP/Ilo/A and PHBV/PCL/PVP/Ilo/A prostheses after 6 months of implantation in the carotid artery in sheep and the contralateral carotid artery.

**Figure 4 ijms-24-08540-f004:**
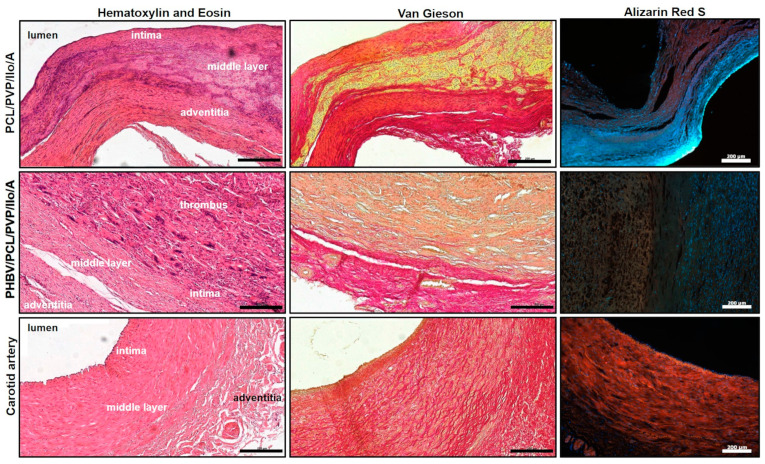
Morphological observation of the explanted PCL/PVP/Ilo/A and PHBV/PCL/PVP/Ilo/A prostheses after six months of implantation into the ovine carotid artery: histological examination with hematoxylin and eosin, Van Gieson and Alizarin Red S staining. Scale bar = 200 µm. Alizarin red-stained samples were visualized using fluorescence microscopy.

**Figure 5 ijms-24-08540-f005:**
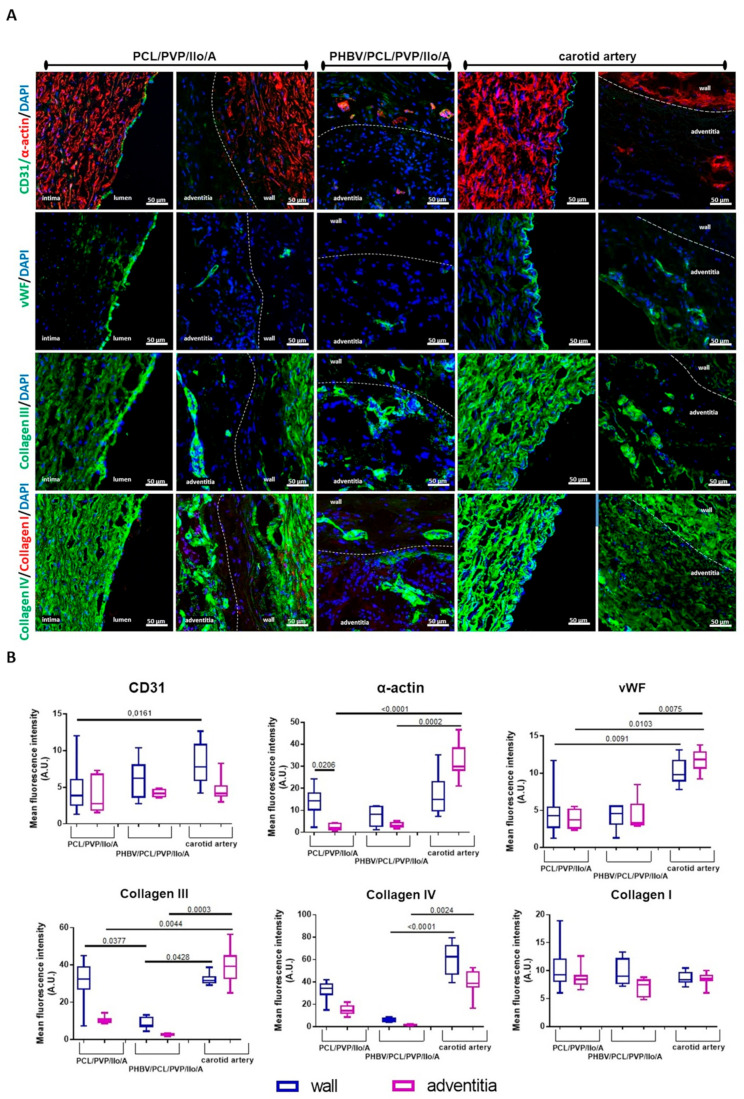
An evaluation of the cell density and collagens amount of explanted PCL/PVP/Ilo/A and PHBV/PCL/PVP/Ilo/A prostheses at six months after implantation. (**A**) Representative immunofluorescent staining of the cross sections of the explanted prostheses: mature endothelial cells stained for CD31 (green) and von Willebrand factor (vWF, green), smooth muscle and other cells containing α-actin (α-actin, red); collagen III (green); collagen I (red) and collagen IV (green). The nuclei of all cells were stained with DAPI (blue). Dashed lines are used to distinguish the boundaries between walls of the prosthesis (wall) and surrounding tissues (adventitia). Scale bar = 50 µm. (**B**) A semi-quantitative analysis of the cell density and collagen deposition was conducted by measuring the mean gray value with ImageJ software, and the data are presented as a box-and-whisker plot. The box represents the median and interquartile range, and the whisker plot represents the minimum and maximum. The statistical analysis was performed using the Kruskal Wallis and Dunn post-test methods.

**Figure 6 ijms-24-08540-f006:**
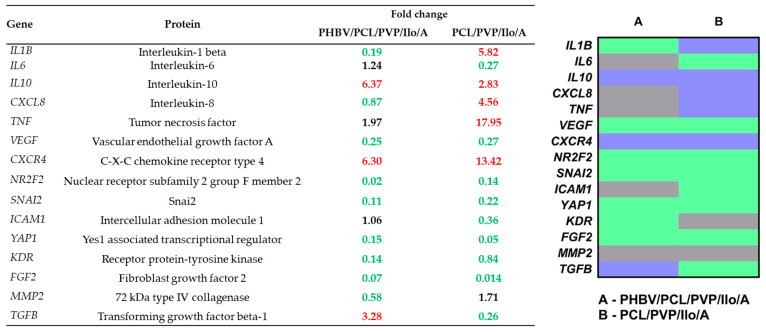
An analysis of the gene expression profiles of the homogenates of PHBV/PCL/PVP/Ilo/A and PCL/PVP/Ilo/A prostheses six months postimplantation compared to the native coronary artery of sheep. Fold-change values indicate a decrease (green) or increase in gene expression (red), or the absence of changes (black). The heat map illustrates the differences in gene expression between the homogenates of the regenerated tissue in PHBV/PCL/PVP/Ilo/A and PCL/PVP/Ilo/A prostheses. Green, grey, and blue colors mean fold change ≤ 0.5, 0.51–1.99, and ≥2, respectively.

**Figure 7 ijms-24-08540-f007:**
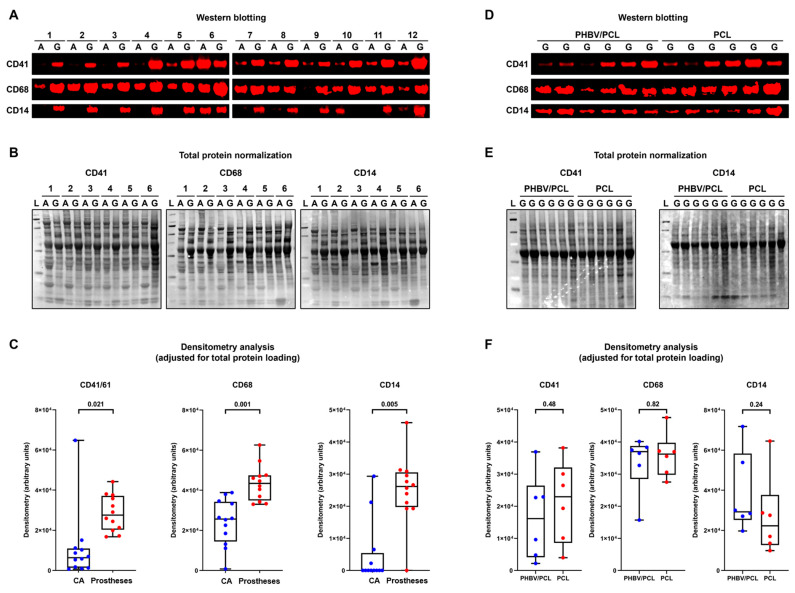
Western blotting of platelet and immune cell markers within the prostheses six months postimplantation as compared to the contralateral ovine carotid arteries. (**A**) Fluorescent Western blotting of intact carotid arteries (A) and prostheses (G) probed for CD41 (**top**), CD68 (**middle**), and CD14 (**bottom**). N = 12 samples per group. (**B**) Total protein staining of the representative membranes (CD41: (**left**); CD68: center; CD14: (**right**)) using Fast Green FCF. (**C**) A semi-quantitative densitometry analysis performed after the pairwise adjustment of the specific signal for total protein staining. The total protein measurement from each prosthesis sample was adjusted for such measurements from the corresponding carotid artery with the following correction of the specific signal quantified in prostheses by Western blotting. Each dot on the plots (blue for carotid arteries and red for prostheses) represents a measurement from one sample (n = 12 measurements per group). Whiskers indicate the range, box bounds indicate the 25–75th percentiles, and center lines indicate the median; p values are provided above boxes, from the Wilcoxon matched-pairs signed-rank test. (**D**) Fluorescent Western blotting of PHBV/PCL/PVP/Ilo/A and PCL/PVP/Ilo/A prostheses probed for CD41 (**top**), CD68 (**middle**), and CD14 (**bottom**). N = 6 prostheses per group. (**E**) Total protein staining of the corresponding membranes (CD41: (**left**); CD14: (**right**)) using Fast Green FCF. (**F**) A semi-quantitative densitometry analysis performed after the adjustment of the specific signal for total protein staining. Each dot on the plots (blue for PHBV/PCL/PVP/Ilo/A and red for PCL/PVP/Ilo/A prostheses) represents a measurement from one sample (n = 6 measurements per group). Whiskers indicate the range, box bounds indicate the 25–75th percentiles, and center lines indicate the median; *p* values are provided above boxes, from the Mann–Whitney U-test.

**Table 1 ijms-24-08540-t001:** Mechanical properties of biodegradable prostheses before and after drug loading in comparison with the sheep carotid artery.

Sample	N	Tensile Strength, MPa Me (25–75%)	Elongation, % Me (25–75%)	Young’s Modulus, MPa Me (25–75%)
Sheep carotid artery	14	1.2 (1.06–1.9)	158.5 (126.0–169.5)	0.49 (0.39–0.66)
PHBV/PCL	7	3.99 (3.71–4.23) ^●^	1438.0 (1403.0–1510.0) ^●^	11.52 (10.66–12.21) ^●^
PCL	7	5.84 (5.56–6.13) ^●^	1391.0 (1350.0–1413.0) ^●^	9.33 (9.23–9.55) ^●^
PHBV/PCL/PVP	7	3.61 (3.23–4.01) ^●^	1202.0 (1120.0–1298.0) *^,●^	19.8 (18.23–20.9) *^,●^
PCL/PVP	7	3.75 (3.48–4.01) *^,●^	1183.0 (1157.0–1215.0) *^,●^	12.86 (11.86–14.06) *^,●^
PHBV/PCL/PVP/Ilo/A	7	3.89 (3.88–3.99) ^●^	1364.0 (1343.0–1393.0) *^,●^	14.55 (13.22–15.24) *^,●^
PCL/PVP/Ilo/A	7	4.02 (3.80–4.18) *^,●^	1454.0 (1433.0–1458.0) *^,●,▲^	10.15 (9.82–10.57) *^,●,▲^

Note: *—*p* < 0.05 in comparison with PHBV/PCL or PCL; •—*p* < 0.05 in comparison with sheep carotid artery; ▲—*p* < 0.05 in comparison with PHBV/PCL/PVP or PCL/PVP. Data are presented as the median (Me) and interquartile range (25–75%).

**Table 2 ijms-24-08540-t002:** Maximum platelet aggregation and hemolysis after blood contact with modified and unmodified biodegradable prostheses.

Sample	Maximum Platelet Aggregation, % Me (25–75%)	Hemolysis, % Me (25–75%)
PHBV/PCL	87.23 (83.95–89.84) *	0.5 (0.0–1.0)
PCL	87.23 (83.27–89.35) *	0.5 (0.0–1.0)
PHBV/PCL/PVP	88.53 (86.59–89.37) *	0.2 (0.0–0.5)
PCL/PVP	90.12 (82.57–90.60) *	0.7 (0.5–1.0)
PHBV/PCL/PVP/Ilo/A	12.18 (11.15–12.24) *^,^**	0.5 (0.0–0.5)
PCL/PVP/Ilo/A	10.7 (10.38–17.23) *^,^**	0.4 (0.0–1.0)
Platelet-rich plasma	74.65 (72.45–75.31)	-

Note: *—*p* < 0.05 in comparison with platelet-rich plasma; **—*p* < 0.05 in comparison with PHBV/PCL or PCL. Data are presented as the median (Me) and interquartile range (25–75%).

**Table 3 ijms-24-08540-t003:** Platelet adhesion on the surfaces of modified and unmodified biodegradable prostheses.

Sample	Type of Platelet, %					Number of Platelets per 1 mm^2^ Me (25–75%)	Deformation Index Me (25–75%)
	I	II	III	IV	V		
PHBV/PCL	7.7	30.8	53.8	7.7	0.0	578.0 (0.0–1349.0)	1.75 (0.0–2.9)
PCL	4.7	46.5	41.9	4.7	2.3	1734.0 (866.9–3179.0) *	2.5 (2.0–2.7) *
PHBV/PCL/PVP	3.0	27.3	45.5	21.2	3.0	1156.0 (0.0–3082.0) *	1.91 (0.0–2.9)
PCL/PVP	12.5	25.5	12.5	50.0	0.0	1728.0 (846.4–3058.0) ^●^	1.9 (0.0–2.8)
PHBV/PCL/PVP/Ilo/A	12.5	62.5	18.8	6.2	0.0	770.6 (0.0–1445.0) ^●^	1.3 (0.0–2.4) ^●^
PCL/PVP/Ilo/A	4.7	6.3	71.8	17.2	0.0	1349.0 (0.0–3275.0)	1.3 (0.0–2.8) ^▲^

Note: *—*p* < 0.05 in comparison with PHBV/PCL; •—*p* < 0.05 in comparison with PHBV/PCL/PVP; ▲—*p* < 0.05 in comparison with PCL/PVP. Data are presented as the median (Me) and interquartile range (25–75%).

**Table 4 ijms-24-08540-t004:** Differentially expressed proteins (fold change ≥ 2.00) in PHBV/PCL/PVP/Ilo/A and PCL/PVP/Ilo/A prostheses. The fold change is calculated as compared with PHBV/PCL/PVP/Ilo/A prostheses. Note the overexpression of 11 contractile proteins (FERM2, TPM1, ACTA, MYLK, ACTH, CNN1, HSPB6, TAGL, CALD1, DEST, and TPM2) in PCL/PVP/Ilo/A as compared with PHBV/PCLPVP/Ilo/A prostheses.

Category	Protein	Description	Uni-Prot	Unique Peptides	PHBV/PCL		PCL		
					Mean Peak Area	n	Mean Peak Area	n	Fold Change
Contraction	FERM2	Fermitin family homolog 2	Q96AC1	41	3412	1	18,019	4	5.28
ECM	CSPG2	Versican core protein	P13611	28	27,186	5	133,182	5	4.90
ECM-BM	PGBM	Basement-membrane-specific heparan sulfate proteoglycan core protein	P98160	74	4743	2	22,393	4	4.72
Contraction	TPM1	Tropomyosin alpha-1 chain	P09493	40	18,892	4	88,790	4	4.70
Contraction	ACTA	Actin, aortic smooth muscle	P62736	52	13,111	1	48,506	4	3.70
Contraction	MYLK	Myosin light chain kinase, smooth muscle	Q15746	55	22,336	3	74,467	5	3.33
ECM	MFAP4	Microfibril-associated glycoprotein 4	P55083	5	46,471	5	146,614	5	3.15
Contraction	ACTH	Actin, gamma-enteric smooth muscle	P63267	51	121,174	6	375,235	6	3.10
Contraction	CNN1	Calponin-1	P51911	44	29,661	4	91,187	4	3.07
Contraction	HSPB6	Heat shock protein beta-6	O14558	15	9473	2	26,632	4	2.81
Contraction	TAGL	Transgelin	Q01995	42	156,077	6	430,898	6	2.76
Contraction	CALD1	Caldesmon	Q05682	67	39,388	6	106,053	6	2.69
Adhesion	CAD13	Cadherin-13	P55290	12	14,034	4	36,982	3	2.64
Contraction	DEST	Destrin	P60981	14	58,347	6	123,228	5	2.11
Inflammation	CRP	C-reactive protein	P02741	16	10,815	3	21,953	4	2.03
Contraction	TPM2	Tropomyosin beta chain	P07951	58	271,460	6	546,685	6	2.01
ECM	CO3A1	Collagen alpha-1(III) chain	P02461	7	21,901	4	43,899	5	2.00

**Table 5 ijms-24-08540-t005:** Electrospinning process parameters.

Parameter	PCL Prosthesis	PHBV/PCL Prosthesis
Voltage, kV	20	22
Flow rate, mL/h	0.5	0.5
Distance between electrodes, mm	150	150
Collector rotation speed, rpm	100	100
Spinneret rate, mm/sec	60	60

PCL—poly-ε-caprolactone; PHBV—poly(3-hydroxybutyrate-co-3-hydroxyvalerate).

**Table 6 ijms-24-08540-t006:** Types of studied vascular prosthesis.

Type	PCL	PHBV/PCL
PVP coated prostheses	PCL/PVP	PHBV/PCL/PVP
Drug coated prostheses	PCL/PVP/Ilo/A	PHBV/PCL/PVP/Ilo/A

PCL—poly-ε-caprolactone; PHBV—poly (3-hydroxybutyrate-co-3-hydroxyvalerate); PVP—polyvinylpyrrolidone; Ilo—iloprost; A—1,5-bis-(4-tetradecyl-1,4-diazoniabicyclo [2.2.2]octan-1-yl)pentane tetrabromide.

**Table 7 ijms-24-08540-t007:** Primer sequences used in the study.

Gene	Forward Primer	Reverse Primer
*B2M*	5′-CCTTCTGTCCCACGCTGAGT-3′	5′-TGGTGCTGCTTAGAGGTCTCG-3′
*ACTB*	5′-AGCAAGAGAGGCATCCTGACC-3′	5′-GGCAGGGGTGTTGAAGGTCT-3′
*GAPDH*	5′-TGGTGAAGGTCGGAGTGAACG-3′	5′-AGGGGTCATTGATGGCAACG-3′
*IL1B*	5′-TGCTGAAGGCTCTCCACCTC-3′	5′-ACCCAAGGCCACAGGAATCTT-3′
*IL6*	5′-TGTCATGGAGTTGCAGAGCAGT-3′	5′-CCAGCATGTCAGTGTGTGTGG-3′
*IL10*	5′-ATGCCACAGGCTGAGAACCA-3′	5′-TCGCAGGGCAGAAAACGATG-3′
*IL12A*	5′-GCAGAAGGCCAGACAAACCC-3′	5′-TGGAAGCCAGGCAACTCTCA-3′
*IL12B*	5′-AGAGCCTGCCCATTGAGGTC-3′	5′-GGTTCTTGGGTGGGTCTGGT-3′
*CXCR4*	5′-CTGGAGAGCAAGCGGTTACCA-3′	5′-ACAGTGGGCAGGAAGATCCG-3′
*CXCL8*	5′-CTTCCAAGCTGGCTGTTGCTC-3′	5′-ATTTGGGGTGGAAAGGTGTGG-3′
*IFNG*	5′-TGAACGGCAGCTCTGAGAAAC-3′	5′-TGGCGACAGGTCATTCATCA-3′
*TNF*	5′-CTTCTGCCTGCTGCACTTCG-3′	5′-TGGCTACAACGTGGGCTACC-3′
*ICAM1*	5′-GTCACGGGGAACAGATTGTAGC-3′	5′-TGAGTTCTTCACCCACAGGCT-3′
*NOS3*	5′-CTTCCGTGGTTGGGCAAAGG-3′	5′-CGTTTCCAGCTCCGTTTGGG-3′
*FGF2*	5′-AGAGCGACCCTCACATCAAACT-3′	5′-TCAGTGCCACATACCAACTGGA-3′
*VEGFA*	5′-GCTTCTGCCGTCCCATTGAG-3′	5′-ATGTGCTGGCTTTGGTGAGG-3′
*TGFB1*	5′-TGAGCCAGAGGCGGACTACT-3′	5′-ACACAGGTTCAGGCACTGCT-3′
*KDR*	5′-ACAGAACCAAGTTAGCCCCATC-3′	5′-TCGCTGGAGTACACAGTGGTG-3′
*MMP2*	5′-ACCCCGCTACGGTTTTCTCG-3′	5′-ATGAGCCAGGAGCCCGTCTT-3′
*NR2F2*	5′-GCAAGCGGTTTGGGACCTT-3′	5′-GGACAGGTAGGAGTGGCAGTTG-3′
*SNAI2*	5′-ACCCTGGTTACTGCAAGGACA-3′	5′-GAGCCCTCAGATTGGACCTG-3′
*YAP1*	5′-TGCTTCGGCAGGAATTAGCTCT-3′	5′-GCTCATGCTCAGTCCGCTGT-3′

## Data Availability

The data presented in this study are available on request from the corresponding author.

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
