# Peer review of "Comparison of the Patency and Regenerative Potential of Biodegradable Vascular Prostheses of Different Polymer Compositions in an Ovine Model"

_ijms, 2023, doi:10.3390/ijms24108540_

Round 1

Reviewer 1 Report

Dear authors

There are some details listed below that the authors should address.I hope that my comment is very useful for the improvement of the article.

(1) In Figure 2, the authors describe the results of PHBV/PCL/PVP/Ilo/A implantation as all the samples were thrombosed, but based on the findings in Figure 3, it is not possible to determine whether it was intimal thickening or white thrombus. Please provide the basis for concluding that they were thrombosed.

(2) The authors describe that PHBV/PCL/PVP/Ilo/A implants showed no indication of endothelial formation, but a large number of neovascular vessels in the middle layer, confirming high gene expression of the receptor CXCR4, which is involved in angiogenesis. In this result, please discuss the mechanism in the increase of angiogenesis-related genes in PHBV molecules.

(3)The authors showed that compared to composite materials made from a mixture of PCL and PHBV, artificial vessels made from PCL alone are less thrombosis, endothelialization, and smooth muscle layer formation, and maintain long-term patency, despite the higher hydrophobicity of the PCL material. These results are very interesting, but no discussion has been given regarding this material exhibiting the above properties. Please discuss in light of previous studies on PCL-based vessels.

(4) In the fabrication of blood vessels by electrospinning, PCL and PHBV are blended to form fibers, but how is it confirmed that each polymer is compatible with each other to form fibers?

(5) In Figure 4, the luminal side of PCL/PVP/Ilo/A appears to be curved. Please show the result of low magnification if you can use supplemental data.

Author Response

Dear Reviewer

Thank you for your insightful comments and suggestions regarding our manuscript. We appreciate the time and effort you have put into reviewing our work and providing valuable feedback. We have carefully considered your comments and have made revisions to our manuscript accordingly. In the following sections, we provide detailed responses to each of your points, explaining the changes we have made and the reasoning behind them.

Dear authors

There are some details listed below that the authors should address. I hope that my comment is very useful for the improvement of the article.

Point 1: In Figure 2, the authors describe the results of PHBV/PCL/PVP/Ilo/A implantation as all the samples were thrombosed, but based on the findings in Figure 3, it is not possible to determine whether it was intimal thickening or white thrombus. Please provide the basis for concluding that they were thrombosed.

Response 1: Blood vessel reconstruction in a sheep model is associated with a high risk of thrombosis. The article presents the results of an assessment of early thrombosis at day 14 post-implantation, which is shown in Figure 2. However, we also examined the implanted vascular prostheses using Doppler ultrasound at day 1 and 5 following implantation, and did not detect blood flow in the lumen of some of the PHBV/PCL/PVP/Ilo/A prostheses (these results have been added to the supplementary materials, Supplementary Figure S3). Fresh thrombi were visualized in these prostheses. Doppler ultrasound images of occluded prostheses 1 day after implantation corresponded to those obtained for the same prostheses at day 14. It's important to note that the development of neointimal thickening with complete occlusion of the prosthesis lumen is a long-term complication that typically occurs over an extended period of time, and as such, 14 days may not be sufficient for its full manifestation. Consequently, the absence of blood flow in prostheses 1, 5, and 14 days after implantation is unlikely to have been caused by neointima hyperplasia and instead suggests that thrombosis was the underlying cause. Thrombosed and patent vascular prostheses were explanted from the carotid artery 6 months after implantation, which allowed for the organization of the thrombus during this period, inevitably impacting its morphology (Figure 4).

We have included this information to the Results.

Point 2: The authors describe that PHBV/PCL/PVP/Ilo/A implants showed no indication of endothelial formation, but a large number of neovascular vessels in the middle layer, confirming high gene expression of the receptor CXCR4, which is involved in angiogenesis. In this result, please discuss the mechanism in the increase of angiogenesis-related genes in PHBV molecules.

Response 2: CXCR4 is the chemokine receptor for CXCL12 (also known as SDF-1) that is expressed by various cell types, including macrophages and T cells. This receptor plays an important role in the homing of progenitor and immune cells [1]. In addition, CXCR4 is involved in various processes including hematopoiesis, organogenesis, as well as vascularization, which was rightly noted by the Reviewer. Our study revealed an increase in CXCR4 gene expression in the remodeled wall of both PCL and PHBV/PCL prostheses. In PCL/PVP/Ilo/A prostheses, CXCR4 gene expression was 2.13 times higher than in PHBV/PCL/PVP/Ilo/A prostheses. We believe that the increase in the expression of genes associated with angiogenesis was not dependent on the type of polymer composition but on the intensity of migration of cells expressing these genes, specifically macrophages involved in polymer bioresorption, into the prosthesis wall. It is likely that in patent PCL/PVP/Ilo/A prostheses, infiltration of various cell types into the wall occurred from both surrounding tissues and the bloodstream, while in thrombosed PHBV/PCL/PVP/Ilo/A prostheses, cell migration into the wall was only possible from the surrounding tissues. Probably, the degree of vascularization of the prosthesis wall can depend on the number of cells that have migrated.

We have added this information to the Discussion.

  1. Bartlett, B.; Ludewick, H.P.; Lee, S.; Verma, S.; Francis, R.J.; Dwivedi, G. Imaging Inflammation in Patients and Animals: Focus on PET Imaging the Vulnerable Plaque. Cells. 2021, 10, 2573. https://doi.org/10.3390/cells10102573

Point 3: The authors showed that compared to composite materials made from a mixture of PCL and PHBV, artificial vessels made from PCL alone are less thrombosis, endothelialization, and smooth muscle layer formation, and maintain long-term patency, despite the higher hydrophobicity of the PCL material. These results are very interesting, but no discussion has been given regarding this material exhibiting the above properties. Please discuss in light of previous studies on PCL-based vessels.

Response 3: The hydrophobicity of PCL limits its use for vascular prostheses due to insufficient hemocompatibility, while PHBV, although more biocompatible, has poor mechanical strength [1]. To address this, we combined PCL and PHBV to compensate for the limitations of each polymer. [2, 3]. In a small animal model, PHBV/PCL prostheses exhibited high patency and efficient remodeling [4]. However, in the sheep model, these prostheses showed poor patency despite antithrombotic surface modification, likely due to their high rigidity. On the other hand, the coating of polyvinylpyrrolidone (PVP) hydrogel containing iloprost on the surface of PCL prostheses resulted in improved patency. PVP reduced the PCL's hydrophobicity, and the lower rigidity of these prostheses provided satisfactory compliance mismatch between the host artery and vascular substitutes. These findings suggest that selecting the appropriate material is crucial to achieving optimal outcomes.

We have added this information to the Discussion.

  1. Kitsuka, T.; Hama, R.; Ulziibayar, A.; Matsuzaki, Y.; Kelly, J.; Shinoka, T. Clinical application for tissue engineering focused on materials. 2022, 10, 1439. https://doi.org/10.3390/biomedicines10061439
  2. Weekes, A.; Bartnikowski, N.; Pinto, N.; Jenkins, J.; Meinert, C.; Klein, T.J. Biofabrication of small diameter tissue-engineered vascular grafts. Acta Biomater. 2022, 138, 92−111. https://doi.org/10.1016/j.actbio.2021.11.012
  3. Rickel, A.P.; Deng, X.; Engebretson, D.; Hong, Z. Electrospun nanofiber scaffold for vascular tissue engineering. Sci. Eng. C Mater. Biol. Appl. 2021, 129, 112373. https://doi.org/10.1016/j.msec.2021.112373
  4. Antonova, L.V.; Sevostyanova, V.V.; Mironov, A.V.; Krivkina, E.O.; Velikanova, E.A.; Matveeva, V.G.; Glushkova, T.V.; Elgudin Ya.L.; Barbarash, L.S. In situ vascular tissue remodeling using biodegradable tubular scaffolds with incorporated growth factors and chemoattractant molecules. Complex Issues of Cardiovascular Diseases. 2018, 7, 25-36. https://doi.org/10.17802/2306-1278-2018-7-2-25-36

Point 4: In the fabrication of blood vessels by electrospinning, PCL and PHBV are blended to form fibers, but how is it confirmed that each polymer is compatible with each other to form fibers?

Response 4: Both PCL and PHBV are soluble in chloroform, enabling the creation of a homogenous solution of a polymer blend suitable for electrospinning. Scanning electron microscopy analysis of electrospun PHBV/PCL vascular prostheses confirmed the formation of smooth polymer fibers, devoid of cracks or beads [1]. Our studies in small and large laboratory animals demonstrated that PHBV/PCL vascular prostheses did not delaminate or disintegrate during or after implantation, even in the long term, suggesting excellent mechanical stability.

  1. Antonova, L.V.; Seifalian, A.M.; Kutikhin, A.G.; Sevostyanova, V.V.; Krivkina, E.O.; Mironov, A.V.; Burago, A.Y.; Velikanova, E.A.; Matveeva, V.G.; Glushkova, T.V.; Sergeeva, E.A.; Vasyukov, G.Y.; Kudryavtseva, Y.A.; Barbarash, O.L.; Barbarash, L.S. Bioabsorbable bypass grafts biofunctionalised with RGD have enhanced biophysical properties and endothelialisation tested in vivo. Front Pharmacol. 2016. 7, 136. https://doi.org/10.3389/fphar.2016.00136

Point 5: In Figure 4, the luminal side of PCL/PVP/Ilo/A appears to be curved. Please show the result of low magnification if you can use supplemental data.

Response 5: The remodeled vascular prostheses had slight variations in wall thickness along their circumference. In thinner areas, histological sections may appear curved. Unfortunately, our microscope's minimum magnification was x50, which meant that the cross-section of the explanted vascular prosthesis with a diameter of 4 mm did not fully fit within the field of view. To address this issue, we included Supplementary Figure S4, which shows the structural features of the wall of the remodeled PCL/PVP/Ilo/A prosthesis in different regions. We correlated histological images with the specific area of the prosthesis examined to provide a more comprehensive understanding of the wall's morphology in various parts of the prosthesis.

Reviewer 2 Report

In the manuscript entitled “Comparison of the patency and regenerative potential of biodegradable vascular prostheses of different polymer compositions in an ovine model”, the authors have extensively investigated drug-loaded electrospun PCL and PHBV/PCL prostheses in a sheep carotid artery interposition model for 6 months. The topic has a great importance in small artery reconstruction and the manuscript is well presenting the results. I have few comments as listed below:

1.       Page 5: please spell out PRP when it’s first mentioned.

2.       Page 14, section 4.2: Did you measure degree of grafting of PVP? Was it similar for PCL and PHBV/PCL?

3.       What is the reason for enhanced mechanical properties when drug was loaded? Did you test mechanical properties of dry prostheses? Would it be more appropriate to test them in wet state?

4.       It was surprising to see PCL demonstrated better regenerative potential than PHBV/PCL in the long-term ovine model. Have you studied in vivo degradation rates of these prostheses? Does PHBV/PCL degrade faster than PCL or are they similar? I think this information should be included in the manuscript.

Author Response

Dear Reviewer

Thank you for taking the time to review our manuscript and for providing constructive feedback. We appreciate your insights and suggestions, which have helped us to improve the quality of our work. In the following sections, we have addressed your comments and provided detailed responses to each point raised.

In the manuscript entitled “Comparison of the patency and regenerative potential of biodegradable vascular prostheses of different polymer compositions in an ovine model”, the authors have extensively investigated drug-loaded electrospun PCL and PHBV/PCL prostheses in a sheep carotid artery interposition model for 6 months. The topic has a great importance in small artery reconstruction and the manuscript is well presenting the results. I have few comments as listed below:

Point 1: Page 5: please spell out PRP when it’s first mentioned.

Response 1: We have deciphered the abbreviation RPR upon its first mention in the article.

Point 2: Did you measure degree of grafting of PVP? Was it similar for PCL and PHBV/PCL?

Response 2: We evaluated the effectiveness of PVP grafting onto the surface of the prostheses using Fourier-transform infrared spectroscopy [1]. In addition, we assessed the success of the modification by detecting the presence of drugs on the prostheses' surface (Figure 1). While we did not evaluate the degree of PVP grafting onto the vascular prostheses in this study, we agree that this analysis could provide further insights into the effectiveness of the modification. Therefore, we plan to include this evaluation in our future studies.

  1. Antonova, L.; Kutikhin, A.; Sevostianova, V.; Lobov, A.; Repkin, E.; Krivkina, E.; Velikanova, E.; Mironov, A.; Mukhamadi-yarov, R.; Senokosova, E.; Khanova, M.; Shishkova, D.; Markova, V.; Barbarash, L. Controlled and synchronised vascular regeneration upon the implantation of iloprost- and cationic amphiphilic drugs-conjugated tissue-engineered vascular grafts into the ovine carotid artery: a proteomics-empowered study. Polymers (Basel). 2022, 23, 5149. https://doi.org/10.3390/polym14235149

Point 3: What is the reason for enhanced mechanical properties when drug was loaded? Did you test mechanical properties of dry prostheses? Would it be more appropriate to test them in wet state?

Response 3: The surface coating of the prostheses with PVP and subsequent grafting with ionizing radiation increased the rigidity of the prostheses. However, immersion of the prostheses in a drug solution and washing the surface removed any remnants of non-polymerized PVP, that decreased the rigidity of the implants. Therefore, we attribute the improvement in the elasticity of the prostheses not to the loading of drugs, but to the removal of unpolymerized PVP.

In this study, we evaluated the mechanical properties of dry samples. We did not see any benefits in testing wet specimens as our previous studies testing wet, uncoated PHBV/PCL or PCL prostheses showed a slight decrease in strength (unpublished data). Furthermore, we observed that PVP-coated prostheses, which become hydrogel when wet, had a slightly reduced Young's modulus compared to dry samples. However, these changes were not statistically significant. Thus, we believe that testing dry samples fully reflects their properties.

Point 4: It was surprising to see PCL demonstrated better regenerative potential than PHBV/PCL in the long-term ovine model. Have you studied in vivo degradation rates of these prostheses? Does PHBV/PCL degrade faster than PCL or are they similar? I think this information should be included in the manuscript.

Response 4: There have been few studies on the patency of biodegradable vascular prostheses in the sheep model. In our previous studies, we evaluated the degradation rate of PCL and PHBV/PCL electrospun samples both in vivo with subcutaneous implantation in rats [1]. The results showed a slow degradation rate of the both polymer compositions, with incomplete degradation observed 12 months after subcutaneous implantation. However, prostheses implanted in the carotid artery of sheep undergo much faster resorption. Within 3.5 months, we observed the formation of aneurysms in the wall of the prostheses due to the rapid bioresorption of the polymer material [2]. Other scientific groups announced the accelerated resorption of polymer products implanted in sheep in parallel with our findings in 2020 [3, 4]. The authors attributed this to the unique characteristics of sheep metabolism.

The rate of biodegradation of polymeric implants depends not only on the properties of the polymer, but also on the shape and morphology of the implant being tested, as well as the animal model used for testing. It is essential to consider all these features when selecting and using polymers for the development and testing of implantable medical devices.

We have included this information in the Discussion.

  1. Nasonova, M.V.; Shishkova, D.K.; Antonova, L.V.; Sevostianova, V.V.; Kudryavtseva, Y.A.; Barbarash, O.L.; Barbarash, L.S. Subcutaneous implantation of poly (3-hydroxybutyrate-co-3-hydroxyvalerate) and poly (ε-caprolactone) scaffolds modified with growth factors. Sovremennye Tehnologii v Medicine. 2017, 9: 7-16. https://doi.org/10.17691/stm2017.9.2.01.
  2. Antonova, L.V.; Krivkina, E.O.; Sevostianova, V.V.; Mironov, A.V.; Rezvova, M.A.; Shabaev, A.R.; Tkachenko, V.O.; Krutitskiy, S.S.; Khanova, M.Y.; Sergeeva, T.Y.; Matveeva, V.G.; Glushkova, T.V.; Kutikhin, A.G.; Mukhamadiyarov, R.A.; Deeva, N.S.; Akentieva, T.N.; Sinitsky, M.Y.; Velikanova, E.A.; Barbarash, L.S. Tissue-engineered carotid artery interposition grafts demonstrate high primary patency and promote vascular tissue regeneration in the ovine model. Polymers (Basel). 2021, 13, 2637. https://doi.org/10.3390/polym13162637.
  3. Matsuzakia, Y.; Iwaki, R.; Reinhardt, J.W.; Chang, Yu-Ch.; Miyamoto, S.; Kelly, J.; Zbindenac, J.; Blum, K.; Mirhaidari, G.; Ulziibayar, A.; Shoji, T.; Breuer, C.K.; Shinoka, T. The effect of pore diameter on neo-tissue formation in electrospun biode-gradable tissue-engineered arterial grafts in a large animal model. Acta Biomaterialia 2020, 115, 176-184. https://doi.org/10.1016/j.actbio.2020.08.011
  4. Fukunishi, T.; Ong, C.S.; Yesantharao, P.; Best, C.A.; Yi, T.; Zhang, H.; Mattson, G.; Boktor, J.; Nelson, K.; Shinoka, T.; Breuer, C.K.; Johnson, J.; Hibino, N. Different degradation rates of nanofiber vascular grafts in small and large animal models. Tissue Eng. Regen. Med. 2020, 14, 203–214. https://doi.org/10.1002/term.2977

Round 2

Reviewer 1 Report

The revised paper has been improved.

Please correct the following points.

- Please recheck Table1 as the line thickness is different.

-Can you provide a high quality image with respect to the MS spectrum in Figure 1? It seems a figure just pasted, e.g. the vertical and horizontal axes are shaded.

Author Response

Dear Reviewer

Thank you for the suggested corrections, they have been incredibly helpful in refining our work. Please find our responses to your suggestions below.

Point 1: Please recheck Table1 as the line thickness is different.

Response 1: We have rechecked Table 1 and found that in the version of the document uploaded to the journal's website, all the lines in the table have the same thickness. It is possible that the thickness of the lines in the table may appear different when the document is opened in a different version of Word or when it is converted to PDF format.

Point 2: Can you provide a high quality image with respect to the MS spectrum in Figure 1? It seems a figure just pasted, e.g. the vertical and horizontal axes are shaded.

Response 2: We have replaced Figure 1 with an image obtained from our laboratory's equipment and added a legend.